# Mitochondrial Genomes of Two Asexual *Trichogramma* (Hymenoptera: Trichogrammatidae) Strains and Comparison with Their Sexual Relatives

**DOI:** 10.3390/insects13060549

**Published:** 2022-06-16

**Authors:** Zhi-Chao Yan, Guang-Yuan Qi, Tian-Yi Yao, Yuan-Xi Li

**Affiliations:** 1Department of Entomology, Nanjing Agricultural University, Nanjing 210095, China; zcyan@njau.edu.cn (Z.-C.Y.); 2020802178@stu.njau.edu.cn (G.-Y.Q.); 2019102089@njau.edu.cn (T.-Y.Y.); 2Key Laboratory of Integrated Pest Management in Crops in Eastern China (Nanjing Agricultural University), Ministry of Agriculture, Nanjing 210095, China

**Keywords:** *Trichogramma*, mitochondrial genome, asexual reproduction, mutation accumulation

## Abstract

**Simple Summary:**

Sexual reproduction is dominant in animals, while asexual lineages are rare and evolutionarily short-lived. However, sexual reproduction has substantial costs, such as male production, inputs to courtship and mating, increased risk of predator exposure, and sexually transmitted diseases. A large body of theories has been proposed to explain the paradox of sex. One favored explanation is that asexuals are more likely to accumulate a greater number of deleterious mutations, known as Muller’s ratchet. *Trichogramma* is a genus of egg parasitoid wasps and is widely used as a biological control agent for agricultural and forest pests. With asexual lineages in at least 16 species, *Trichogramma* provides an excellent model to investigate the causes and consequences of asexual reproduction. In this study, we sequenced and assembled two asexual *Trichogramma* mitogenomes, representing two divergent origins of asexual reproduction. The asexual *T. pretiosum* is induced by the endosymbiont *Wolbachia*, while *T. cacoeciae* presumably originates from interspecific hybridization. To test Muller’s ratchet hypothesis, we compared these two asexual mitogenomes with their sexual relatives and found no association between asexual reproduction and mutation accumulation. This study provides a basis for further investigation into mitochondrial evolution and asexual reproduction in *Trichogramma*.

**Abstract:**

Despite its substantial costs, sexual reproduction dominates in animals. One popular explanation for the paradox of sex is that asexual reproduction is more likely to accumulate deleterious mutations than sexual reproduction. To test this hypothesis, we compared the mitogenomes of two asexual wasp strains, *Trichogramma cacoeciae* and *T. pretiosum,* to their sexual relatives. These two asexual strains represent two different transition mechanisms in *Trichogramma* from sexual to asexual reproduction. Asexual *T. pretiosum* is induced by *Wolbachia*, while *T. cacoeciae* presumably originated from interspecific hybridization. We sequenced and assembled complete mitochondrial genomes of asexual *T. cacoeciae* and *T. pretiosum*. Compared to four sexual relatives, we found no evidence of higher mutation accumulation in asexual *Trichogramma* mitogenomes than in their sexual relatives. We also did not detect any relaxed selection in asexual *Trichogramma* mitogenomes. In contrast, the intensified selection was detected in Nad1 and Nad4 of the asexual *T. pretiosum* mitogenome, suggesting more purifying selection. In summary, no higher mitochondrial mutation accumulation was detected in these two asexual *Trichogramma* strains. This study provides a basis for further investigating mitochondrial evolution and asexual reproduction in *Trichogramma*.

## 1. Introduction

Sex is one of the most important evolutionary innovations [1]. Compared to asexual reproduction, sexual reproduction has substantial costs, such as male production, inputs to courtship and mating, and increased risks of predator exposure and sexually transmitted diseases [2,3]. Despite these apparent costs, sex is the overwhelmingly dominant mode of reproduction in animals. Asexual animals are rare, although they are distributed among various evolutionary taxa, e.g., insects, mites, rotifers, reptiles, amphibians, and fishes [4]. Asexual lineages are also mostly of recent origins from sexual lineages, occupying the terminals on the tree of life. This suggests that asexual reproduction is an evolutionary dead-end that eventually leads to extinction [4]. Why does sexual reproduction dominate in animals? It remains a mystery and one of the greatest puzzles in biology [1].

Many hypotheses have been proposed to explain the paradox of sex [1]. One class of favored explanations is that sexual reproduction can remove deleterious mutations more effectively through recombination and selection [5]. On the other hand, as all loci are linked, asexual reproduction is more likely to accumulate deleterious mutations in a stochastic (Muller’s ratchet) or deterministic (Kondrashov’s hatchet) manner and leads to population decay and extinction [5]. The prediction of accelerated mutation accumulation in asexual lineages can also be extended to nonrecombining mitochondrial genes [6,7]. This is because mitogenomes cannot be segregated from nuclear loci in asexual organisms, similar to the inability to recombine between nuclear loci. The complete linkage with the entire nucleus leads to less efficient purifying selection and the accumulation of more deleterious mutations [6,7].

The validity of these theories can be tested by comparison between asexual lineages and their sexual relatives. Several studies have shown excess mutation accumulation in both mitochondrial and nuclear genomes of asexual lineages [8,9,10,11,12]. For example, in the microcrustacean *Daphnia pulex*, mitochondrial protein-coding genes accumulate deleterious mutations in asexual lineages at four times the rate of their sexual relatives [11]. Relaxed selection was detected in these asexual lineages [6]. However, a nonnegligible number of studies have indicated that there is no relationship between asexual reproduction and excess mutation accumulation [13,14,15,16,17,18]. Even a recent study showed that asexual lineages have more efficient purifying selection than their sexual relatives in oribatid mites [19].

In insects, asexual reproduction is rare but widely distributed among both Hemimetabola and Holometabola [20,21]. *Trichogramma* is a hymenopteran genus of egg parasitoid wasps and is widely used as a biological control agent for agricultural and forest pests [22]. As only female parasitoid wasps can parasitize pest hosts, asexual reproduction is particularly significant in biocontrol to increase the production of pest-killing female parasitoids [23]. In *Trichogramma*, several asexual strains have been reported [24,25]. In most of them, asexual reproduction is caused by the endosymbiont *Wolbachia*, and antibiotics or high temperatures can restore normal sexual reproduction [26,27]. There are also asexual *Trichogramma* strains, which cannot be cured by antibiotics or high temperatures [26]. For example, cytogenetic studies revealed that asexual *T. cacoeciae* is maintained through an apomictic mechanism, where meiotic cells divide only once followed by the expulsion of a single polar body [28]. It is entirely different from the mechanism of *Wolbachia*-induced asexual reproduction through gamete duplication [29]. Consistent with these mechanisms, *Wolbachia*-induced asexual *Trichogramma* females are completely homozygous [29], while asexual *T. cacoeciae* individuals are highly heterozygous, presumably resulting from interspecific hybridization [28]. Thus, *Trichogramma* provides an excellent model to investigate the causes and consequences of asexual reproduction with at least two different modes of transition from sexual to asexual reproduction.

In this study, we sequenced the mitochondrial genomes of two asexual *Trichogramma* strains. The asexual *T. pretiosum* is induced by *Wolbachia* [18], while the asexual *T. cacoeciae* presumably originates from interspecific hybridization [28]. We reported the full-length mitogenomes of these two asexual *Trichogramma* strains and compared their features with those of mitogenomes from four sexual *Trichogramma*. Specifically, we tested whether these *Trichogramma* asexual mitogenomes show more mutation accumulation than those from their sexual relatives.

## 2. Materials and Methods

### 2.1. Sample Preparation and DNA Extraction

*Trichogramma cacoeciae* was retrieved from the Beijing Academy of Agriculture and Forestry Sciences, and *T. pretiosum* was retrieved from Hainan University. As previously described [30], *Trichogramma* wasps were reared in the Insectary of Nanjing Agricultural University using UV-irradiated *Corcyra cephalonica* eggs as hosts. *Trichogramma* specimens were stored in 100% ethanol at −20 °C before DNA extraction. According to the manufacturer’s protocol, DNA from a single *Trichogramma* wasp was extracted using the Wizard SV Genomic DNA Purification System Kit (Promega Beijing Biotech, Beijing, China).

### 2.2. Mitochondrial Genome Sequencing

Initial PCRs were conducted using previously reported Cox1 universal primers [31] (FP1 and RP1; Appendix A). PCR products were Sanger-sequenced (Tsingke Biotech, Nanjing, China). Based on the sequenced Cox1, taxon-specific primers (FP2 and RP2 for *T. cacoeciae*; FP3 and RP3 for *T. pretiosum*; Appendix A) for long PCR were designed using Primer Premier 5 [32]. Long PCRs for the near full-length mitogenomes were conducted using Tks Gflex DNA Polymerase (Takara Bio Inc., Kusatsu City, Japan). The PCR cycling settings were as follows: activation at 98 °C for 2 min, followed by 35 cycles of denaturing at 98 °C for 10 s, annealing at 55 °C for 10 s, and extension at 68 for 10 min. For Illumina sequencing of long PCR products, libraries were prepared using the TruSeq Nano DNA LT Prep Kit (Illumina, San Diego, CA, USA) according to the manufacturer’s protocol. For *T. cacoeciae*, the library was sequenced using Illumina HiSeq X Ten with PE150 mode by Biozeron Co. Ltd., Shanghai, China. For *T. pretiosum*, the library was sequenced using an Illumina NovaSeq 6000 with PE150 mode by Annoroad Gene Technology Co., Ltd., Beijing, China.

### 2.3. Mitochondrial Genome Assembly

For *T. cacoeciae*, the full-length mitogenome was assembled by Geneious v9.1.4 [33] using Illumina reads and the Sanger-sequenced Cox1 sequence. For *T. pretiosum*, Illumina reads were first assembled using MitoZ v2.4 [34], and the Cox1 sequence was then merged into the assembly using Geneious v9.1.4. Since there was still a gap between Cox1 and Cox3, additional primers, FP4 and RP4, were designed using Primer Premier 5 to fill this gap (Figure 1; Appendix A). We also noticed an erroneous insertion of 14 amino acids within Nad4l in the *T. pretiosum* mitogenome assembly. A pair of primers, FP5 and RP5, were designed for amplification and Sanger-sequencing of this insertion region (Figure 1; Appendix A). This erroneous insertion was corrected by the Sanger-sequencing result. PCRs in this section were performed using Rapid Taq Master Mix (Vazyme, Nanjing, China). The PCR cycling settings were as follows: activation for 3 min at 95 °C, followed by 35 cycles of 30 s at 95 °C, 30 s at 54 °C, and 1 min at 68 °C. The final cycle was followed by an extension of 10 min at 68 °C.

### 2.4. Gene Annotation and Nucleotide Diversity Analyses

Gene annotation was performed using the MITOS web service [35]. tRNAs were also predicted by the ARWEN web service [36]. All mitochondrial genes were manually checked based on mitogenome alignments. Nucleotide diversity (Pi) was calculated for 13 mitochondrial protein-coding genes (PCGs) and two mitochondrial ribosomal RNAs (rRNA) using DNASP v6 [37]. Window slide nucleotide diversity was also estimated for 7 aligned *Trichogramma* mitogenomes using DNASP v6. The window size was set as 200 bp, and the step was set as 20 bp. Sites with gaps were excluded as the default setting in DNASP.

### 2.5. Phylogenetic Analyses

Mitochondrial proteins were aligned using Mafft v7.487 with LINSI mode [38]. Alignments were filtered and then translated back to codon sequences using trimAl v1.4.rev15 [39]. Ribosomal RNAs were aligned using the R-Coffee web service by considering secondary structure [40]. Alignments of rRNAs were also filtered using trimAl. The first and second codon positions of PCGs were extracted and concatenated with rRNAs. The phylogenetic tree was constructed using IQ-Tree v2.2.0 [41]. The best-fit model was automatically selected by MolderFinder [42], which was built in IQ-Tree. *Megaphragma amalphitanum* was set as the outgroup. The other two gene sets were also analyzed, resulting in the same phylogenetic topology. They are (1) all three codon positions of 13 PCGs and (2) 13 PCGs concatenated with two rRNAs. The solved phylogeny was used as a fixed topology for evolutionary rate estimation. Substitution models were set as mtART for mitochondrial proteins and GTR for mitochondrial rRNAs. Synonymous substitution rates (Ks) and nonsynonymous substitution rates (Ka) were estimated using PAML v4.9 [43].

### 2.6. Relaxed Selection Test

Relaxed selections were tested using RELAX in Hyphy v2.5.36 [6]. *Trichogramma cacoeciae* and *T. pretiosum* terminal branches were used as the test. All the other terminal and internal branches within the *Trichogramma* genus were set as the reference. A relaxation or intensification parameter (K) significantly less than 1 indicates relaxed selection, while a K value significantly greater than 1 indicates intensified selection. *p* values were corrected for multiple tests using the Benjamini and Hochberg method.

## 3. Results

### 3.1. Features of Two Asexual Mitochondrial Genomes

Combining both Illumina and Sanger sequencing data, we assembled two complete mitochondrial genomes from asexual strains of *T. cacoeciae* and *T. pretiosum*, respectively (Figure 1). The complete mitochondrial genome of *T. cacoeciae* is 16,034 bp, with an AT% of 84.9%, and that of *T. pretiosum* is 16,227 bp, with an AT% of 85.2%. Both the length and AT% of asexual *T. cacoeciae* and *T. pretiosum* mitogenomes are within the range of four sexual *Trichogramma* mitogenomes. The control regions of *T. cacoeciae* and *T. pretiosum* are 756 bp and 669 bp, respectively. The AT% of the control region in *T. cacoeciae* is 87.6%, which is the lowest AT% among the six sequenced *Trichogramma* species. The AT% of the control region in *T. pretiosum* is 91.6%, which is the highest among these six.

All 37 typical mitochondrial genes were identified in both the *T. cacoeciae* and *T. pretiosum* mitogenomes (Figure 1; Table 1). Thirteen PCGs, 22 tRNAs, and two rRNAs were annotated on each mitogenome. All tRNAs have typical cloverleaf structures, except trnS2. Mitochondrial trnS1 lacks the dihydrouridine (DHU) arm in all six *Trichogramma* mitogenomes, which is common in metazoans [44]. Gene lengths are similar among the six *Trichogramma* mitogenomes, and there are no significant differences among the different *Trichogramma* mitogenomes (Table 1; Kruskal–Wallis test: *χ*^2^ = 0.168, *df* = 5, *p* = 1.00). Extensive gene order rearrangement was previously reported in *Trichogramma* mitogenomes, even when compared to *M. amalphitanum*, a close relative in the same family Trichogrammatidae [45]. However, within the *Trichogramma* genus, the gene orders of the six sequenced *Trichogramma* mitogenomes are all the same. Window slide nucleotide diversity (Pi) was also estimated, showing variance across the whole mitochondrial genome (Figure 2). The highest window slide Pi is 0.20667 in atp8, while the lowest is 0.01733 in rrnL. For the average Pi of mitochondrial PCGs, atp8 has the highest value of 0.136, while cox2 has the lowest value of 0.057. The average Pi of rrnL and rrnS is 0.080 and 0.068, respectively.

### 3.2. Comparison of Substitution Rates between Asexual and Sexual Mitogenomes

We first estimated substitution distances of all six *Trichogramma* to the *Trichogramma* common ancestor (TCA), which reflect the accumulated substitutions after divergence from their common ancestor (Figure 3). For the 13 concatenated mitochondrial proteins, the distance of *T. ostriniae* is the longest to the TCA, while that of *T. japonicum* is the shortest (Figure 3A). For the two concatenated mitochondrial rRNAs, the distance of T. chilonis is the longest, while that of T. dendrolimi is the shortest (Figure 3B). There is no difference between distances to the TCA when comparing asexual and sexual mitogenomes (Mann–Whitney test: for proteins, *W* = 4, *p* = 1; for rRNA, *W* = 3, *p* = 0.8). We then tested whether this tendency is the same among individual genes (Figure 3C). There are variations among different genes (Kruskal–Wallis test: *χ*^2^ = 68.11, *df* = 14, *p* = 4.2 × 10^−9^). Cox1 shows the shortest substitution distance to the TCA with an average of 0.0149, while atp8 shows the highest with an average of 0.1618. However, there was no difference among the different *Trichogramma* mitogenomes (Kruskal–Wallis test: *χ*^2^ = 4.12, *df* = 5, *p* = 0.53).

We further estimated synonymous (Ks) and nonsynonymous substitution rates (Ka) using concatenated PCGs (Figure 4). For six *Trichogramma* species, Ka/Ks values range from 0.005864 to 0.009482, suggesting that mitochondrial PCGs are under strong purifying selection. The Ka/Ks values are 0.008530 on the *T. cacoeciae* branch and 0.007323 on the *T. pretiosum* branch, showing no significant difference from the Ka/Ks values of their sexual relatives (Mann–Whitney test: *W* = 5, *p* = 0.8). We also tested relaxed selection on asexual branches. By setting *T. cacoeciae* and *T. pretiosum* terminal branches as the test and other branches as the reference, we did not obtain any parameter K significantly less than 1 (Table 2), suggesting no relaxed selection on asexual branches. In contrast, there was a K for asexual branches on Nad4, which was significantly greater than 1 (K = 5.9, LR = 186.52, *q* = 0), suggesting intensified selection. Similarly, by setting asexual *T. cacoeciae* or *T. pretiosum* as the test separately, no signals of relaxed selection were detected (Appendix A), while intensified selection was detected on Nad1 and Nad4 for *T. pretiosum* (for Nad1: K = 2.65, LR = 11.57, *q* = 0.007; for Nad4: K = 5.67, LR = 132.56, *q* = 0).

## 4. Discussion

We assembled full-length mitochondrial genomes of two asexual *Trichogramma* strains and compared them with four reported mitogenomes of sexual *Trichogramma*. Crucially, we found no significant differences between asexual and sexual mitogenomes. Asexual and sexual *Trichogramma* mitogenomes show the same gene order and similar gene lengths. Compared to their sexual relatives, there was no excess mutation accumulation or relaxed selection detected in the asexual *Trichogramma* mitogenomes.

Comparisons of asexual lineages with their sexual relatives provide a promising avenue to answer the paradox of sex. However, such comparisons are difficult, as factors other than sex also affect mutation accumulation, such as mutation rate, recombination rate, generation time, population size, and the ecological environment [46]. These factors may mask the association between asexual reproduction and mutation accumulation. More than 200 species have been reported in the genus *Trichogramma*, and asexual reproduction has been found in at least 16 species [24]. One consideration of our study is that our sampling is very limited. Sparse sampling can result in asexual branches containing most of the sexual stages and a small proportion of the asexual stages, thereby underestimating the difference between asexual and sexual *Trichogramma*. Additionally, we have no information on the exact time of origin of these two asexual strains. Previously, Lindsey et al. (2018) reported no evidence of degradation in the asexual nuclear genome by comparing asexual and sexual *T. pretiosum* strains [18]. One possible explanation is that the transition to asexual reproduction is too recent for a detectable acceleration of mutation accumulation.

Another influencing factor is the selection of gene sets. Several studies have demonstrated excess accumulation of mitochondrial mutations in asexuals than their sexual relatives [6,9,11,12]. However, the mitochondrial genome may be less sensitive than the nuclear genome in detecting the association between asexual reproduction and mutation accumulation [7]. For example, compared to its sexual relatives, the asexual parasitoid wasp *Diachasma muliebre* has a higher mutation accumulation in the nuclear genome but not in the mitochondrial genome [47]. This may be because mitochondria are pivotal in energy production and are usually under strong purifying selection [48]. Additionally, the mitochondrial genome is maternally transmitted and lacks recombination in both sexual and asexual reproduction [48,49]. Moreover, some genes are more likely to decay than others, e.g., genes associated with sexual traits. For instance, Kraaijeveld et al. (2016) identified 16 putative degraded nuclear-encoded genes, the majority of which (15/16) are associated with male traits, and one is expressed in the female-specific tissue spermathecae [50].

In addition, different mechanisms of asexual reproduction may result in different evolutionary outcomes. Asexual reproduction in *T. pretiosum* is induced by *Wolbachia* through gamete duplication [29]. Even after removing *Wolbachia* by antibiotics, some *Wolbachia*-induced asexual strains cannot revert to normal sexual reproduction because of the loss of sexual functions in either male or female adults [24,51]. However, during early *Wolbachia* infection in a population, infected *Trichogramma* females can still be fertilized by uninfected males [29]. Gamete duplication is suppressed, and *Wolbachia* does not interfere with the normal sexual development of fertilized eggs [29]. This unique mechanism can reduce the rate of mutation accumulation. In contrast, asexual *T. cacoeciae* is presumed to originate from interspecific hybridization [28]. *Trichogramma cacoeciae* individuals are highly heterozygous and remain diploid by an apomictic mechanism [28]. It is still largely unknown how different mechanisms of asexual reproduction influence mutation accumulation.

It is also possible that asexual organisms have compensatory mechanisms to avoid deleterious mutation accumulation. For example, asexual oribatid mites show more effective purifying selection than their sexual relatives [19]. The method by which these ancient asexual mites escape mutation accumulation is still unknown. Some molecular mechanisms have been reported to slow down mutation accumulation, e.g., gene conversion in the human Y chromosome and DNA repair in mitochondria [52,53]. Asexual organisms can also recruit similar mechanisms to prevent mutation accumulation. In the asexual *T. pretiosum* mitogenome, we found intensified selection on the Nad1 and Nad4 genes, suggesting more purifying selection. Whether this is widely associated with *Wolbachia*-induced asexual reproduction needs further investigation.

In summary, we reported two mitochondrial genomes from asexual *T. cacoeciae* and *T. pretiosum* strains and found no evidence of excess mutation accumulation in these two asexual mitogenomes when compared to sexual *Trichogramma* mitogenomes. This study provides a basis for further investigating mitochondrial evolution and asexual reproduction in *Trichogramma*.

## Figures and Tables

**Figure 1 insects-13-00549-f001:**
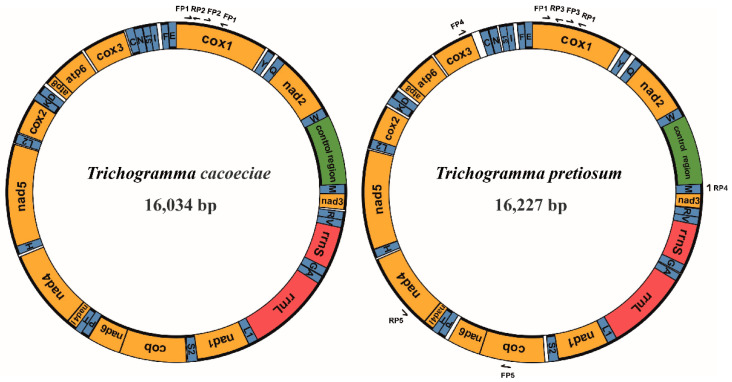
Structures of *Trichogramma cacoeciae* and *T. pretiosum* mitochondrial genomes. Yellow indicates mitochondrial protein-coding genes. Red indicates mitochondrial ribosomal RNAs. Blue indicates mitochondrial tRNAs. Green indicates the control region, which is also known as the D-loop region, major noncoding region, or A + T-rich region. Genes on the inner loop are encoded on the major strand, while those on the outer loop are on the minor strand. Arrows indicate primers used for mitochondrial PCR amplification. FP: forward primer; RP: reverse primer.

**Figure 2 insects-13-00549-f002:**
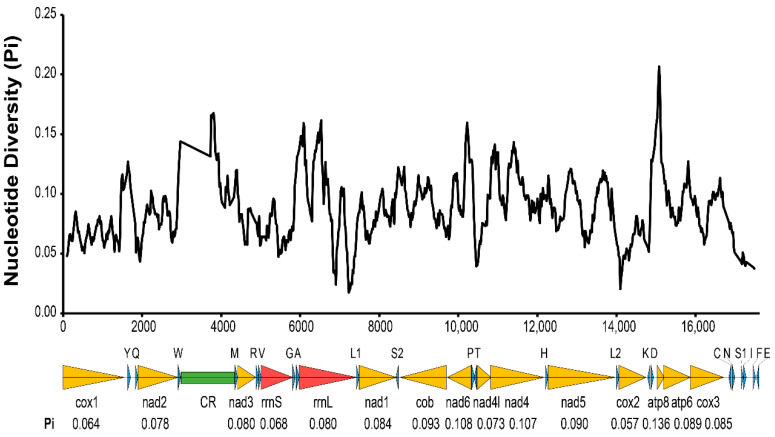
Nucleotide diversity (Pi) of six *Trichogramma* mitochondrial genomes. The x-axis indicates the position in the alignment of six mitochondrial genomes. The black wave line indicates the average nucleotide diversity of sliding windows. The window size was 200 bp with a step size of 20 bp. The arrows below the graph indicate 37 mitochondrial genes. Yellow indicates mitochondrial protein-coding genes. Red indicates mitochondrial ribosomal RNAs. Blue indicates mitochondrial tRNA. Green indicates the control region. The average Pi of protein-coding genes and ribosomal RNAs are labeled under each gene name.

**Figure 3 insects-13-00549-f003:**
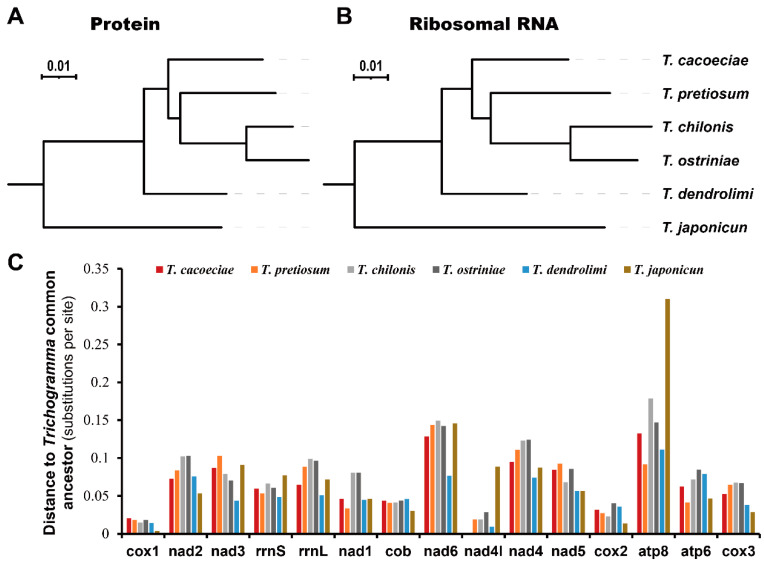
Comparison of substitution rates among *Trichogramma* species. Substitution rates were estimated using (**A**) concatenated 13 mitochondrial proteins or (**B**) concatenated two ribosomal RNAs. (**C**) Distances to *Trichogramma* common ancestor for individual mitochondrial proteins or ribosomal RNAs. Genes are ordered by their locations on mitochondrial genomes.

**Figure 4 insects-13-00549-f004:**
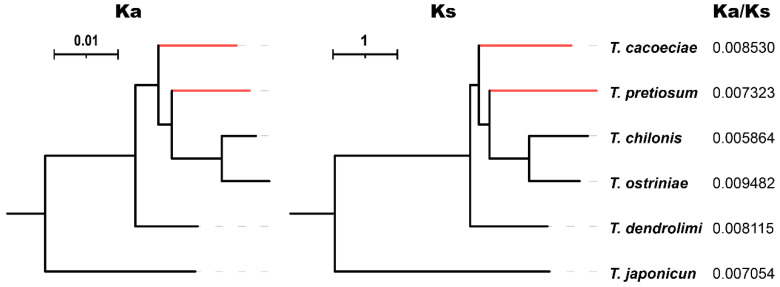
Comparison of Ka/Ks ratios among *Trichogramma* species. The nonsynonymous substitution rate Ka and synonymous substitution rate Ks were estimated using 13 concatenated mitochondrial protein-coding genes. Red indicates asexual branches, which were tested for relaxed selection.

**Table 1 insects-13-00549-t001:** Summary of mitochondrial genes in *Trichogramma cacoeciae* and *T. pretiosum*.

Gene	Strand	*T. cacoeciae*		*T. pretiosum*		*Tchi*	*Tden*	*Tjap*	*Tost*
		Start/Stop	Length (bp)	Start/Stop	Length (bp)	Length (bp)	Length (bp)	Length (bp)	Length (bp)
cox1	+	ATG/TAA	1536	ATG/TAA	1548	1536	1536	1536	1536
trnY	+		68		66	66	67	67	66
trnQ	+		68		70	68	68	68	68
nad2	+	ATA/TAA	1014	ATA/TAA	1014	1014	1014	1014	1014
trnW	+		66		66	67	66	66	67
trnM	+		65		65	64	65	68	66
nad3	+	ATA/TAA	360	ATA/TAA	360	360	363	363	360
trnR	+		64		63	63	63	65	63
trnV	+		67		67	68	68	66	68
rrnS	+		770		784	777	780	749	774
trnG	+		66		67	66	66	65	67
trnA	+		63		64	63	65	65	64
rrnL	+		1388		1417	1393	1393	1367	1362
trnL1	+		68		70	70	66	65	70
nad1	+	ATA/TAA	939	ATA/TAA	939	936	936	939	936
trnS2	−		64		64	64	65	65	64
cob	−	ATG/TAA	1140	ATG/TAA	1143	1140	1140	1140	1140
nad6	−	ATG/TAA	573	ATG/TAA	573	579	594	573	573
trnP	+		66		65	66	65	66	65
trnT	−		64		64	64	64	65	64
nad4l	+	ATT/TAG	288	ATT/TAG	288	288	288	288	288
nad4	+	ATG/TAA	1344	ATG/TAA	1344	1344	1344	1344	1344
trnH	+		64		67	63	65	67	63
nad5	+	ATT/TAA	1689	ATT/TAA	1683	1683	1689	1686	1692
trnL2	+		66		66	68	66	66	66
cox2	+	ATT/TAG	681	ATT/TAA	681	681	681	681	681
trnK	−		70		72	70	70	70	70
trnD	+		66		68	66	67	66	66
atp8	+	ATT/TAA	168	ATA/TAA	159	168	168	159	159
atp6	+	ATA/TAA	672	ATA/TAA	672	681	681	675	675
cox3	+	ATG/TAA	792	ATG/TAA	792	792	831	792	792
trnC	−		69		69	69	69	69	68
trnN	+		66		67	66	66	66	62
trnS1	+		60		60	60	60	59	60
trnI	+		67		67	67	67	67	67
trnF	+		66		65	65	65	64	65
trnE	−		66		67	66	66	67	66

Tchi: *T. chilonis*; Tden: *T. dendrolimi*; Tjap: *T. japonicum*; Tost: *T. ostriniae*.

**Table 2 insects-13-00549-t002:** Relaxed selection test of asexual branches, *T. cacoeciae*, and *T. pretiosum*, compared to sexual branches.

Gene	K	LR	*p*	*q*
atp6	1.13	0.15	0.700	0.899
atp8	0.69	0.41	0.520	0.899
cob	1.00	0.00	1.000	1.000
cox1	1.05	0.04	0.835	0.899
cox2	1.99	2.44	0.118	0.413
cox3	1.07	0.06	0.799	0.899
nad1	1.72	5.83	0.016	0.112
nad2	1.20	0.31	0.576	0.899
nad3	1.55	0.64	0.425	0.899
nad4	5.90	186.52	0	0
nad4l	14.98	0.22	0.639	0.899
nad5	0.77	3.74	0.053	0.247
nad6	0.93	0.08	0.771	0.899
13PCGs	1.11	1.17	0.280	0.784

Terminal branches of *T. cacoeciae* and *T. pretiosum* were used as the test. Other terminal and internal branches were set as the reference. A K value significantly less than 1 indicates relaxed selection, while a K value significantly greater than 1 indicates intensified selection. *p* values were corrected for multiple tests using the Benjamini and Hochberg method. K: relaxation or intensification parameter; LR: likelihood ratio; 13 PCGs: concatenated 13 mitochondrial protein-coding genes.

## Data Availability

The mitochondrial genomes of *T. chilonis* (MW789210.1), *T. dendrolimi* (KU836507.1), *T. japonicum* (NC_039534.1), *T. ostriniae* (NC_039535.1), and *M. amalphitanum* (NC_028196.1) were downloaded from NCBI. The mitochondrial genomes of *T. pretiosum* and *T. cacoeciae* have been deposited in GenBank (ON209198 and ON209199).

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
