# Peer review of "Mitochondrial Genomes of Two Asexual Trichogramma (Hymenoptera: Trichogrammatidae) Strains and Comparison with Their Sexual Relatives"

_insects, 2022, doi:10.3390/insects13060549_

Round 1
Reviewer 1 Report
Dear Dr. Li,
I have read an updated version of your manuscript entitled "Mitochondrial genomes of two asexual Trichogramma (Hymenoptera: Trichogrammatidae) strains and comparison with their sexual relatives". As far as I can see, the quality of your paper is substantially improved now, and I therefore believe that your manuscript could be published in Insects after a careful revision of the language, preferably by a native English speaker.
Author Response
RESPONSE: We thank the reviewer’s careful reading and constructive suggestions. We have carefully read through the entire manuscript and revised the language. Revisions are tracked in the tracking version.
Reviewer 2 Report
Thanks to the opportunity, the authors have addressed my main concern. The article is fine and can be accepted for publication.
Once more I recommend an improvement in the figures and graphics.
All the best.
Author Response
RESPONSE: We thank the reviewer’s careful reading and constructive suggestions. All the figures have been updated with higher resolution. Separate figures are also provided in the file named “Figures.zip”. Each figure was set at the resolution of 600 dpi.
This manuscript is a resubmission of an earlier submission. The following is a list of the peer review reports and author responses from that submission.
Round 1
Reviewer 1 Report
In this manuscript, the authors compared two asexual Trichogramma mitochondrial genomes. They selected single line for each species, which make it impossible to evaluate the extent of intraspecific variations and compare the two species using different asexual reproduction mechanism. Unfortunately, the results presented in this manuscript just showed that no higher mitochondrial mutation accumulation was detected in these two species. It is not surprising because the maternal inheritance mechanism of mitochondria is consensus between asexual and sexual reproduction. I guess that their objective is associated with breeding and quality control of egg parasitoid wasps as a biotic pesticide. If so, they should investigate another lines maintained independently.
Reviewer 2 Report
Dear Dr. Li,
I have carefully read your manuscript entitled "Mitochondrial genomes of two asexual Trichogramma (Hymenoptera: Trichogrammatidae) strains and comparison with their sexual relatives". I can see that your paper contains new information on mitochondrial genomes of two asexual members of the genus Trichogramma with different mechanisms of thelytoky, as well as an important conclusion about the substantial lack of mutation accumulation within the mitochondrial genomes of both species. I therefore believe that your manuscript could be published in Insects. However, I have a few suggestions which could improve the manuscript. First, I would recommend citing the comprehensive review by Gokhman and Kuznetsova (https://doi.org/10.1111/jzs.12183), which lists arrhenotokous and thelytokous forms in holometabolous insects, including Hymenoptera, to put your study into a broader context. Second, it would probably be useful to mention that Lindsey et al. (2018), who studied the nuclear genome of Trichogramma pretiosum, also did not find any strong evidence for genome degradation in the asexual vs. sexual form of this species.
Reviewer 3 Report
Dear all, sorry for my delay but I was reading carefully and try to reply on the manuscript more than 2 times.
I do agree that the manuscript describe the mitochondrial genomes properly. Nice figure (can be improved resolution and graphics) and all necessary data about the genomes were discussed.
My main concern is with the methods and hypotheses tested. I do not believe that Mueller`s reached can be properly tested/adressed comparing mitogenomes.
Mitochondria are in general under selective pressure and do not follow restrict Medelian inheritance.
Thus, in my opinion the obvious results would be absence of correlation due spurious analysis.
I would like to see what the authors think about.
One suggestion is basically describe the mitochondrial genomes and emphasized another questions as gene orders, etc...